# Outcome of Root Canal Treatments Provided by Endodontic Postgraduate Students. A Retrospective Study

**DOI:** 10.3390/jcm9061994

**Published:** 2020-06-25

**Authors:** Carmen Llena, Teodora Nicolescu, Salvadora Perez, Silvia Gonzalez de Pereda, Ana Gonzalez, Iris Alarcon, Angela Monzo, José Luis Sanz, Maria Melo, Leopoldo Forner

**Affiliations:** Departament of Stomatology, University of Valencia (Valencia-Spain), Gascó Oliag 1, 46010 Valencia, Spain; teo_mdf_15@hotmail.com (T.N.); doraperez11@hotmail.es (S.P.); dragonzalezdepereda@gmail.com (S.G.d.P.); anagonzaliz@gmail.com (A.G.); iris_3lin@hotmail.com (I.A.); angelamonzo@hotmail.com (A.M.); jsanzalex96@gmail.com (J.L.S.); maria.melo.alminana@gmail.com (M.M.); forner@uv.es (L.F.)

**Keywords:** apical periodontitis, intraoperative factors, postoperative factors, preoperative factors, root canal treatment, success

## Abstract

The aim of this study was to assess the preoperative, intraoperative, and postoperative factors that influenced complete periapical healing in teeth that underwent primary root canal treatment (RCT), in patients treated by postgraduate students in endodontics. Factors were retrieved and compared with the periapical status during the follow-up visit. Healing was considered as the absence of clinical and radiological symptoms. Variables significantly associated by the chi-squared test were included in a logistic regression model (LRM). Preoperative factors associated with healing were: American Society of Anesthesiology (ASA) status (*p* = 0.01); the absence of preoperative pain (*p* = 0.04); positive response to pulp tests; when the RCT cause was caries, pain, abscess, or sinus tract; probing depth <4 mm; the absence of mobility; absence or <4 mm periapical lesion (*p* < 0.01). In the LRM, the factors included were: absence or <4 mm periapical lesion; probing depths <4 mm; RCT caused by caries, pain, abscess, or sinus tract; the tooth was not a bridge abutment. Postoperative factors were: teeth with direct restoration; teeth that did not act as a support for a fixed prosthetic restoration; the favorable condition of the coronal restoration (*p* < 0.01). In the LRM, only the status of the coronal restoration was included. Preoperative conditions and the adequate fit of the coronal restoration influenced the outcome of RCT.

## 1. Introduction

Apical periodontitis is an acute or chronic inflammatory process of polymicrobial origin which reaches the periodontium through the root canal system [1]. It is characterized by exhibiting high cytokine and inflammatory mediator levels, which trigger a periapical inflammatory response through the activation of the innate immune system [2]. Any modification in the innate immune system can alter this response.

Ideally, the process of apical periodontitis healing will result in a complete disappearance of any radiological alterations in the periradicular tissues. This reparative process involves the neoformation of connective tissue at the lesion site, which will be substituted by bone tissue [3].

Various clinical situations and therapeutic factors may influence periapical lesion healing. The control of the infection in the dentin–pulp complex by means of the correct debridement and cleaning of the root canals, together with a functional apical seal, are factors to be considered, since they will favor the conditions for healing. Nonetheless, the underlying local and/or systemic conditions of the patients may affect the outcome of root canal treatment (RCT) [4].

According to the recommendations of the European Society of Endodontology (ESE), the outcome of RCT has to be evaluated at least 1 year after the intervention. A favorable result will include: the absence of pain, inflammation, and sinus tracts, together with an absence of radiological signs of periapical pathology. If there is radiographic evidence for the persistence of the initial periapical lesion, with a small reduction or no change in size, the outcome will be considered as uncertain and will require a follow-up of at least 4 years. If the lesion persists for longer than 4 years, or signs of apical resorption can be observed, the outcome of the treatment will be considered as a failure, requiring a retreatment [5].

Many studies have reported high success rates (complete healing) after RCT [6]. The majority of them base their evaluation on the clinical characteristics and radiographic evolution of the status of the periapical tissue [7]. The success rates of primary RCTs ranged from 68% to 85%, according to a systematic review [8], and from 70% to 86% after retreatment, when strict criteria for the complete absence of periapical radiolucency were applied [9]. When only clinical aspects were considered, the survival rates ranged from 86–93% [9,10]. The substantial differences between the reported success and survival rates of RCT can be attributed to the methodological heterogeneity of the studies.

The factors involved in the repair of apical periodontitis are varied, including not only preoperative and intraoperative factors, but also postoperative factors, such as tooth function and load, and the quality of the coronal restoration [11].

Accordingly, the aim of the present study was to assess the preoperative, intraoperative, and postoperative factors that influenced periapical lesion healing rates in teeth that underwent primary RCTs, after a follow-up periods of 1 to 6 years, in patients treated by postgraduate students in endodontics.

## 2. Methodology

### 2.1. Experimental Procedure

The present retrospective observational study was approved by the ethics committee from the Universitat de València, with the file number: H1445365577359.

Clinical data from the patients were introduced to an electronic database with a restricted access code. Once introduced, they were anonymized to allow a blinded analysis.

The target population was patients within an age range of 18–81 years, who underwent primary RCT carried out by students from the postgraduate program in endodontics of the Universitat de València (Valencia, Spain) under the supervision of a clinician specialized in endodontic practice. By means of the clinical history data, patients who met the aforementioned criteria within the period of 2013–2018 were selected to evaluate their eligibility for the study.

The inclusion criteria were: teeth which underwent a primary RCT, at least 1 year before their clinical examination. Patients who received medication which could alter bone metabolism, namely immunosuppressors, corticosteroids, or antiresorptives, were excluded. Teeth which suffered from periodontal disease, which had undergone a previous RCT, which were extracted after the primary RCT for non-endodontic reasons, which did not present a preoperatory periapical radiograph that allowed a correct identification of the radiographic apex, or those for which pre/intra/postoperative information was missing were also excluded.

Different treatment protocols were selected depending on the anatomical, radiographic, and clinical characteristics of the teeth. Nonetheless, all of the treatments were carried out following the standards from the European Society of Endodontology [5].

A preoperative X-ray was obtained using a phosphor plate X-ray system (VistaScan 2+; Dürr Dental, Bietigheim-Bissingen, Germany) and paralleling system rings (XCP RINN; Dentsply Sirona, York, PE, USA).

The treatments were performed under local anesthesia and rubber dam isolation. After making the access cavity, root canals were located using a straight probe and patency was achieved using a #10 K manual file (FKG dentaire SA, La Chaux de Fonds, Switzerland). Root canal working length was determined using an electronic apex locator (Root ZX; J Morita Co, Tustin, CA, USA) and confirmed with a periapical radiograph (Vistascan 2+; Durr dental SE, Bietigheim-Bissingen, Germany). Once confirmed, root canals were shaped using either manual or rotary files to different degrees, depending on the characteristics of the root canal systems and the clinicians’ preferences. The minimum recommended apical instrumentation was equivalent to a #30 caliber. If the diameter was wider initially, no further widening was recommended.

Throughout the instrumentation process, 3 mL of 2.5% or 5.25% sodium hypochlorite (Dentaflux, Madrid, Spain) were used as an irrigant solution between files, carried in 27 gauge side-cut open-ended needles (Monoject luer lock syringe; Sherwood medical, St. Louis, MO, USA). After instrumentation, the following final irrigation sequence was performed: 2.5% saline solution, 17% EDTA (Clarben SA, Madrid, Spain), 2.5% saline solution, and lab-prepared 2% chlorhexidine. Irrigant solutions were activated using an ultrasonic device (EMS; Electro Medical Systems SA, Nyon, Switzerland/P5; Satelec Acteon group, Merignac, France) with a low power setting and a #15 K file (Dentsply Maillefer, Ballaigues, Switzerland).

Calcium hydroxide (Ultracal, Ultradent Products INC., South Jordan, UT, USA) was the standard interappointment medicament, if necessary.

Root canal filling was carried out using gutta percha and epoxy resin-based sealer AH Plus (Dentsply Maillefer, Switzerland), by means of the lateral condensation technique, modified lateral condensation technique using preheated manual spreaders, or vertical compaction. In some of the latter cases, Bioroot RCS (Septodont, Louisville, CO, USA) was used as a root canal sealer. In all cases, a periapical radiograph was obtained after the RCT.

Coronal restorations were placed using composite from various manufacturers, supported by a fiber post (from different manufacturers), if necessary, for retention. The maximum period between RCTs and the placement of coronal restorations was 2 weeks. Cavit (3M Espe, Madrid, Spain) was used as a temporary restoration material.

### 2.2. Clinical Follow-Up Evaluation

The clinical history records of patients were retrieved. Patients who fulfilled the inclusion criteria were contacted via telephone and appointed for a follow-up evaluation by one of the investigators. All patients were previously informed about the objectives of the study and were asked to sign an informed consent. From the clinical history records, preoperative, intraoperative, and postoperative data were extracted (Table 1 and Table 2).

In the follow-up evaluation, patients were asked about perceived signs and symptoms, if any, specifically the presence of pain, swelling/inflammation, sinus tracts, or mobility. Then, the evaluation proceeded with a clinical and radiological exploration of the tooth or teeth eligible for the study, using the same technique described above. If the teeth were extracted, they were excluded from the analysis, since the cause for extraction was difficult to determine.

### 2.3. Healing Criteria

Both clinical and X-ray criteria were assessed. The analyzed variables are presented in Table 3.

For the patients who presented clinical symptoms, even if radiological periodontal involvement was not detected and the appearance of the root canal treatment was correct in terms of the root canal filling and length, X-ray images were carried out in different projections to detect the potential cause, and they were considered as non-healing.

The preoperative, intraoperative, and postoperative X-rays were evaluated by two independent examiners to ensure intra- and inter-observer concordance. Both examiners were clinicians specialized in endodontic practice, different from those who carried out the treatment. The size of periapical radiolucency was determined using the measurement tool from VistaScan software (Dürr Dental, Bietigheim-Bissingen, Germany).

RCTs were considered as favorable when the gutta percha exhibited an adequate compaction (without defects or spaces inside the root canal) and extended up to 2 mm from the radiographic apex. Postoperative periodontal state was assigned to one of three categories: (1) healing, when there was no evidence of periodontal ligament affectation, (2) minor or no reduction in lesion size, and (3) increase in lesion size. In the event of disagreements between the examiners or uncertainties in the evaluation of the periodontal state, the cases were excluded from the analysis.

### 2.4. Statistical Analysis

The statistical analysis of the results was carried out using the SPSS v26.0 statistical package (SPSS, Inc., Chicago, IL, USA). Cohen’s kappa coefficients were calculated to assess both intra- and inter-observer agreement on X-ray examination. Good agreement was considered as >0.8, substantial as 0.61–0.8, and moderate as 0.4–0.6.

Preoperative, intraoperative, and postoperative variables (categorical variables) were associated with the response variable, which was divided into two categories: healing/no healing (absence/presence of periapical radiolucency; respectively), through the chi-squared (χ^2^) test. The follow-up period (continuous variable) was associated with the response variable by means of the *t*-test. Variables which were found to be associated with healing were included in a logistic regression model. In all cases, statistical significance was considered at *p* < 0.05.

## 3. Results

The mean age of the included patients was 55.5 ± 16.43, ranging from 18–81 years old, with a gender distribution of 43.1% male and 56.9% female patients. Different preoperative conditions were present, as described in Table 1. None of them showed a radiographic open apex.

During the assessed period of time (2013–2018), a total of 1227 primary RCTs were performed. All of the corresponding clinical records were screened, and after applying the inclusion criteria, 820 teeth (66.82%) from 796 patients resulted as eligible. From the 820 teeth, 182 were excluded by not attending the follow-up evaluations. An additional 33 teeth were also excluded due to extraction. A total of 605 were assessed, from which 20 were excluded due to doubts regarding the state of the tooth in the final evaluation. Finally, 585 teeth were included for analysis. Of those, 65.8% underwent a follow-up period of 1–2 years, 24.4% 2–4 years, and 9.9% 4–6 years. A flow diagram representing the follow-up of patients which met the inclusion criteria, reasons for exclusion, and the total number of evaluated teeth is presented in Figure 1.

Intra- and inter-observer agreement was assessed with regards to the radiographic evaluation. Intra-observer kappa values were 0.81 and 0.83, respectively, and the inter-observer kappa value was 0.82.

The patients in whom the initial periapical radiolucency was healed after RCT were of a mean age of 54.41 ± 16.12 years old, and those in whom the area did not heal or was only partially healed were 58.01 ± 14.16 years old (*p* = 0.01). Data regarding the preoperative state are presented in the first column of Table 1. Values are distributed by gender, teeth, American Society of Anesthesiology (ASA) status, and preoperative variables. In the third column, the absolute values and percentages of teeth which achieved complete healing, and their association with preoperative variables (by means of the chi-squared test), are presented. ASA I patients presented a higher percentage of complete healing than ASA II patients (*p* = 0.01), as did teeth which presented a positive response to vitality tests in comparison to non-vital teeth (*p* < 0.01). The presence of an apical radiolucency of <4 mm, or its absence, was associated with a significantly higher percentage of complete healing compared to the presence of a >4 mm periapical radiolucency. Teeth referring prolonged spontaneous or stimuli-induced pain, or pain upon percussion, presented a higher percentage of complete healing than those without pain (*p* = 0.04). When the cause for RCT was caries, pain, abscess, or sinus tract, the percentage of teeth with complete healing was significantly higher than that obtained when the cause for RCT was a traumatism, accidental pulpar exposure, or other causes (*p* < 0.001). Lastly, a probing depth of <4 mm, as well as the absence of mobility, were associated with a higher number of teeth with complete healing (*p* < 0.001).

The assessed intraoperative factors are presented in Table 2. Data are presented in a similar manner as in Table 1. Variables were related to instrumentation, filling, treatment sessions, and intraoperative complications, and none of them were significantly associated with periapical lesion healing (*p* > 0.05).

Regarding postoperative factors (Table 2), variables related to the post-endodontic coronal restoration (immediate and after follow-up evaluation), and teeth function, were assessed. Healing percentages were significantly higher in teeth with direct composite restorations than teeth with crowns or those which acted as prosthetic bridges (*p* < 0.001). Teeth in which restorations were in a favorable state upon the follow-up evaluation, whether direct restorations or crowns, presented a higher percentage of complete healing than those with maladjusted or leaked restorations (*p* < 0.001). 91.1% of the teeth assessed from 1–2 years posterior to the RCT exhibited a complete healing of the initial periapical radiolucency. Healing percentages were lower in longer follow-up periods, although the differences were not significant (*p* = 0.48). The mean follow-up period for teeth which showed complete healing (*n* = 530) was 23.97 months (CI 95%: 22.03–25.92). No significant association between follow-up time and healing was found (*p* = 0.29).

The clinical and radiographic variables considered in the follow-up evaluation, and the absolute values and percentages of treated teeth which presented them, are illustrated in Table 3. The total number of teeth with complete apical lesion healing was 530 (90.6%), and 39 teeth (6.7%) presented a minor or no reduction in the initial periapical radiolucency. In the remaining teeth, the periapical radiolucency showed an increased size compared to their initial state, or a new periapical lesion appeared.

The variables which were significantly associated with healing (using a chi-squared test), were introduced to a logistic regression model using the Wald test (Table 4). In the analysis, block 0 indicated that the model presented a 90.6% probability of verisimilitude (*p* < 0.01). For block 5 of the model, ROA’s efficiency score statistic indicated that there was a statistically significant improvement in the prediction of the probability of the occurrence of the categories from the dependent variable (healing/no healing), with a value of *p* < 0.001. The Nagelkerke R^2^ value indicated that the model would explain 39% of the dependent variable. A Hosmer and Lemoshowen test indicated that for block 5, the model had a probability of success of 92% (χ^2^ = 7.51 (5 df) *p* = 0.18). Altogether, statistical results indicate a favorable adjustment of the model. The preoperative variables which were finally included in the model were: periapical radiolucency (for instance, a 1–4 mm periapical radiolucency multiplies the risk of no healing by 0.23 (Exp(B)), this factor being the one which influences the evolution of the case less), periodontal probing depth, and the cause of treatment. As postoperative factors, the type of restoration and a favorable restoration condition at follow-up were included.

## 4. Discussion

The objective of the present retrospective study was to assess the factors that influenced periapical lesion healing in teeth that underwent primary RCTs, after follow-up periods of 1–6 years, in patients treated by postgraduate students in endodontics.

Retrospective studies have been widely used among the literature to evaluate the extent to which certain factors may have an influence on periapical lesion healing after RCT. Factors selected for such an evaluation may be more or less restrictive. In general terms, various studies base their analyses on one to three factors [3,12,13,14,15,16,17], while others assess the influence of multiple factors, taking into account preoperative, intraoperative, and postoperative factors [7,18,19,20,21,22,23,24]. The latter is the case of the present study. Usually, this kind of study describes outcomes from RCT performed by general practitioners or by endodontists [2,3,4,6,7,8,9]. In this case, our study is focused on cases treated by postgraduate students. This allows for the evaluation of the postgraduate teaching–learning process, as well as the efficacy of the endodontic clinical activity.

The retrospective nature of these studies, however, involves a series of limitations. Most importantly, ensuring the reliability of the information collected from records is not always possible [25]. In the present study, to minimize this limitation, strict inclusion criteria were applied for the analysis, excluding records in which there was any missing information or it was insufficiently clear. Extracted teeth were not included, due to the impossibility of evaluation in the follow-up visits.

Likewise, it should be highlighted that follow-up rates in longitudinal studies are around 50%, which also entails a risk that could alter the outcome values and percentages [26]. In the present study, from the cases which met the inclusion criteria, only 22.19% could not be evaluated due to patients not attending the follow-up evaluations. As shown in Figure 1, the number of patients decreases as the follow-up period increases. Nonetheless, to evaluate the extent to which factors influence periapical lesion healing, the importance of follow-up visits must be emphasized. In this study, a follow-up of at least 1 year was established, following the recommendations of the ESE [5], which could be extended up to 4 years if the lesion had not healed. According to the criteria proposed by Ng et al. [8], if strict criteria for periapical lesion healing are applied, follow-up periods of 3 years should be established. However, in the present retrospective study, 91.1% of the primary RCTs evaluated within 1–2 years healed completely, maintaining similar values up to 4 years of follow-up (90.9%). A study carried out by Torabinejad et al. in 2018 [27] on 120 roots with cone beam computed tomography (CBCT) reported no association between follow-up time and the evolution of periapical lesions. This difference may be attributed to the different diagnostic method used in their study and the longer follow-up period (2–15 years).

With regards to the method used to evaluate healing, the majority of studies use conventional or paralleled digital periapical radiographic images [20]. In a meta-analysis performed by Pak et al. in 2012 [28], the heterogeneity of the radiographic diagnostic methods of the studies assessing periapical lesions is highlighted, as well as the difficulty in assessing minor changes. In the present study, paralleled digital radiographs were used, and in the event of any disagreements between the examiners or uncertainties in the evaluation of the periodontal state, the cases were excluded from the analysis. For this reason, a total of 20 teeth were excluded.

Currently, CBCT has demonstrated up to twice the precision to evaluate the evolution of periapical radiolucencies [16,27,29,30,31]. However, following the As Low As Reasonably Achievable (ALARA) principles, the systematic use of CBCT for the evaluation and follow-up of periapical lesion healing may not be justified. According to the ESE, CBCT should only be used when a precise diagnosis cannot be achieved by means of a conventional radiographic study [32].

### 4.1. Influence of Preoperative Factors on Periapical Lesion Healing

Studies in the field mainly include basic data regarding the age, gender, and general state of health of the patients, expressed using the ASA status classification system. With reference to age, the majority of studies report no association with the treatment outcome. A systematic review of 24 articles performed by Shakiba et al. demonstrated that age does not influence RCT success [33]. However, Landys et al. [7] found a higher survival rate among younger patients. In the present study, the *t*-test analysis revealed that the age of patients whose teeth achieved a complete healing was significantly lower than those with no healing. Nonetheless, the age variable was not included in the multivariate model, and thus cannot be considered as a prognostic factor of periapical lesion healing.

In this study, no significant association was found between gender and periapical lesion healing. Some studies reported such an association, especially in favor of women [7,34], while others found no association [35]. Authors attribute this tendency to the fact that women show a greater interest in their health, attend check-up visits more frequently, and tend to receive more root canal treatments [36].

Regarding the patients’ health state (registered by means of the ASA status), a significantly higher percentage of teeth with complete healing was reported from the ASA I group. However, this factor was not included among the prognostic factors in the logistic regression model. Alterations in the general health state of patients may exert an influence on periapical lesion healing after RCT [37]. In the present study, no specific pathologies have been distinguished. However, there is evidence for the relationship between periapical lesion healing and diabetes [23,38,39]. Such an association becomes less evident with regards to cardiovascular diseases or HIV infection [23,37].

The type of tooth, root canal anatomy, and number of root canals have been described as factors influencing periapical lesion healing [29,34,35,40,41]. Multi-rooted teeth and teeth with a more complex anatomy potentially present a lower probability of healing [42,43], so we considered three groups of teeth according to the difficulty of the case, following the classification from the American Association of Endodontists (AAE). In the present study, no association was found between periapical lesion healing and the type of tooth. However, all of the teeth which presented a new periapical lesion in the follow-up evaluation (*n* = 9, 1.5%) were molars. The complex anatomy found among molars could explain why, despite the correct appearance of the RCT, there may have been a non-favorable outcome.

With reference to the pulp condition prior to the RCT, Friedman et al. reported, in part I of the Toronto studies, a significant association between pulp vitality and RCT success [40]. A systematic review of 63 studies performed by Ng et al. [8] concluded that RCT success rate decreases, with or without an initial periapical lesion, in non-vital teeth. Likewise, a meta-analysis carried out by Kojima et al. [44] also reported lower success rates among non-vital teeth when compared to vital teeth. In the present study, the same tendency was confirmed by means of the chi-squared test. Nonetheless, this variable was not included in the logistic regression analysis.

Similarly, the presence of initial apical periodontitis prior to RCT was significantly associated with a reduced number of teeth with complete healing. A greater periapical lesion size was also negatively associated with healing, both in the chi-squared test and the logistic regression model. These results coincide with the available literature, not only regarding the presence of the apical lesion [34,40,41], but also periapical lesion size [45].

Another factor related to the state of the periapical lesion is its exacerbation, which manifests as inflammation or a sinus tract. Their presence indicates a periapical infection of a persistent and chronic nature, and have been associated with a worse prognosis in various studies [20,21,38]. In this study, no significant association was found between healing and the presence of inflammation or a sinus tract, although teeth without these signs did present a higher percentage of healing (91.5% vs. 86.1%).

Regarding the presence of pain prior to the RCT, a significant inverse association with healing was found. Patients who referred no preoperative pain presented a significantly lower percentage of complete healing compared to those with spontaneous pain, stimuli-induced pain, or pain upon percussion by means of the chi-squared test. However, this variable was not included in the logistic regression model. The absence of spontaneous or stimuli-induced pain can be indicative of pulp necrosis, which may be associated with chronic apical periodontitis. This would justify the lower percentage of healing shown. In fact, 52.9% of the teeth with a >4 mm periapical radiolucency, and 29.6% of teeth with a periapical radiolucency of 1–4 mm, did not refer pain, while 59.7% of patients who referred pain did not present a periapical radiolucency. Differences in microbial composition and the presence of a more structured biofilm, together with the difficulty of achieving an effective disinfection of the root canal system in teeth with pulp necrosis or chronic apical periodontitis, could justify these results.

Lastly, with reference to the RCT cause, a higher percentage of healing was significantly associated with cases in which the treatment cause was caries, pain, or inflammation, both in the chi-squared test and in the logistic regression analysis. These results may be related to other factors that may influence the evolution of RCT, such as the presence of fissures in the case of traumatisms or prosthesis [9,11].

### 4.2. Influence of Intraoperative Factors on Periapical Lesion Healing

In terms of intraoperative factors, evidence suggests that the results are influenced by the operator’s experience [3,11,29,46]. Particularly in teeth with a more complex anatomy, such as molars, healing percentages appear to be higher if treated by clinicians specialized in endodontic practice [3,42]. In the present study, all of the RCTs were carried out by postgraduate students, who are expected to have greater knowledge than a general practitioner, but less experience than a clinician specialized in endodontic practice.

Another factor which was taken into consideration was the number of visits for RCT. Manfredi et al., in a systematic review of 25 trials, concluded that RCTs performed in a single visit may be accompanied by pain, without influencing the outcome of the treatment [47]. Other studies have reported that healing rates are similar regardless of the number of visits [12,40,48,49]. In this study, no significant association was found between the number of treatment sessions and periapical lesion healing.

Regarding the influence of intraoperative complications in RCT outcomes, such as instrument fractures, perforations, apical stops or seats, or a failure to achieve patency, no clear association is found among various studies [22,40,43,50]. Others, however, reported that they negatively influence the RCT outcome [34,38]. Similarly, in the last phases of the Toronto study, it was described that the presence of intraoperative complications acts as a predictive factor of RCT outcome [34]. Iqbal et al., in a university study on a population from Saudi Arabia, determined that the complications which influenced healing the least were perforations (5.5%) and fractured instruments (6.6%) [42]. In the present study, intraoperative complications only occurred in 6.8% of the cases, of which perforations and apical transportations were the most frequent. No significant association was found between the presence of intraoperative complications and apical lesion healing.

The quality and extension of the root canal filling has been extensively studied. It is generally considered as “adequate” if the gutta percha filling extends 0–2 mm from the radiographic apex. The available literature demonstrates that a correct root canal filling, in terms of extension and compaction, will lead to a favorable long-term prognosis [6,9,14]. Among “inadequate” filling, in terms of extension, short filling presents a higher probability of success than over-filling [9]. From the results of the present study, it should be highlighted that, despite the fact that the percentage of healing was higher in teeth in which root canal filling was correct, the difference was not significant. However, among teeth which did not present a periapical radiolucency in the follow-up evaluation, 79.2% presented a correct filling. In the cases where a new periapical lesion appeared, 70% of them presented an incorrect filling. Therefore, although the extension of the root canal filling could not be considered as a prognostic factor in the present study, from a clinical perspective, a correct root canal filling may favor apical healing.

The instrumentation technique used in RCT, whether manual or rotary, may also influence its outcome. In a systematic review performed by Ng et al. in 2008 [9], the authors inferred that there were insufficient data regarding the influence of this factor. Similarly, in a study by Fleming et al. [12], no significant differences were found in terms of the endodontic technique used. On the other hand, Connert et al. [51], in a study on a German population, described that the type of instrumentation did influence the association with apical periodontitis. Likewise, in one of the phases of the Toronto study [34], it was reported that the type of root canal preparation influenced teeth with initial apical periodontitis. Altogether, a lack of consensus can be appreciated among the literature with regards to this factor. In the present study, a higher percentage of healing was found among teeth which were prepared using a rotary technique when compared to the manual technique (94.4% vs. 87.3%), but this difference was not significant. Results were different if teeth with and without an initial periapical lesion were assessed independently. In the cases in which an initial periapical radiolucency was present, those prepared manually healed in 73.6% of the cases, while those prepared using a rotary technique healed in 69.4% of the cases (*p* = 0.23). However, those which did not present an initial periapical radiolucency healed in 74.3% using a manual preparation, and 96% using a rotary technique (*p* < 0.01). Thus, in cases with no initial periapical lesion, the instrumentation technique appears to influence RCT outcome.

### 4.3. Influence of Postoperative Factors on Periapical Lesion Healing

Regarding the type of restoration placed after RCT, the percentage of teeth with complete healing was higher in those with a direct restoration when compared to a crown, or teeth which acted as a support for a fixed prosthesis. This difference was significant both in the bivariate and multivariate analysis. However, no association was found with reference to the presence or absence of antagonist teeth. Among the literature, some authors found higher success rates in teeth restored with crowns [7,20], while others did not find an association [11]. With regards to the prosthetic load, a systematic review performed by Ng et al. [9] reported that the RCT success rate decreased if the teeth posteriorly acted as a support for a fixed prosthesis, as did the present study.

An extensively studied factor among the literature is the state of the coronal restoration on follow-up visits. Various studies have demonstrated that an unfavorable state of the restoration negatively influences the outcome of RCT [45,52]. It has also been reported that teeth with a favorable coronal restoration have a 1.82 times higher probability of success than an unfavorable restoration [8], as well as a higher probability of periapical lesion healing [53]. In the present study, the percentage of teeth with periapical lesion healing was significantly higher in teeth with a favorable coronal restoration in the follow-up visits.

## 5. Conclusions

Preoperative factors which influenced healing after primary RCT were the absence of or <4 mm X-ray periapical lesions; probing depths <4 mm; RCT caused by caries, pain, or inflammation, and when the tooth was not a bridge abutment. As a postoperative factor, only the favorable condition of the restoration in the follow-up visit influenced healing. None of the intraoperative factors influenced healing after the primary RCT performed by endodontic postgraduate students.

## Figures and Tables

**Figure 1 jcm-09-01994-f001:**
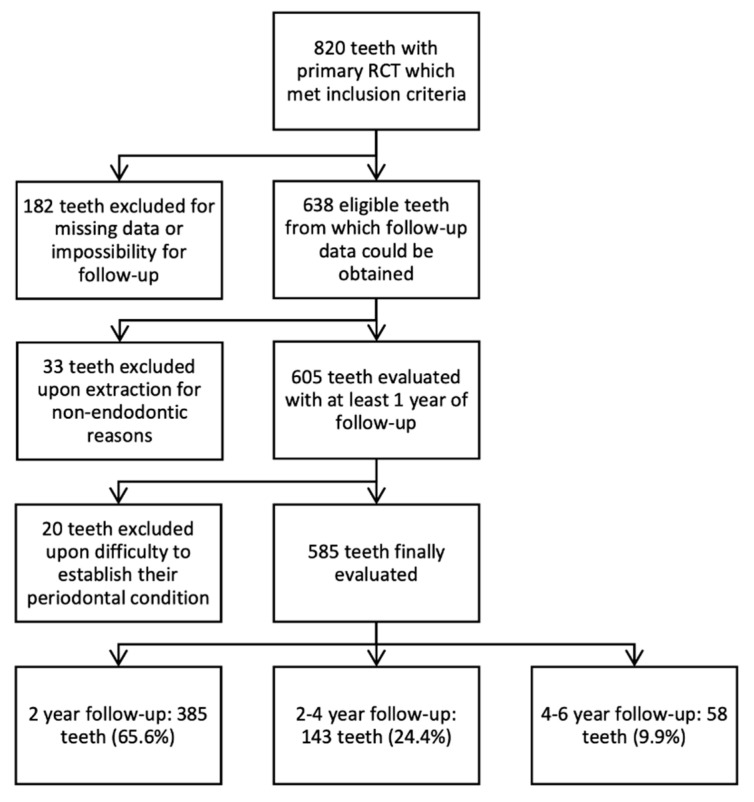
Flow diagram representing the follow-up of patients which met the inclusion criteria.

**Table 1 jcm-09-01994-t001:** Registered preoperative factors.

Preoperative Factors	Teeth with RCT	Complete Healing	*p*
Men	252 (43.1%)	234 (92.9%)	0.06
Women	333 (56.9%)	298 (88.9%)	
Anterior teeth and premolars	256 (43.8%)	227 (88.7%)	0.06
First molars	178 (30.4%)	158 (88.8%)	
Second and third molars	151 (25.8%)	145 (96%)	
ASA I	413 (70.6%)	382 (92.5%)	0.01
ASA II	172 (29.4%	148 (86%)	
Vital teeth	365 (62.3%)	349 (95.6%)	0.00
Non-vital teeth	220 (37.7%)	181 (82.3%)	
Absence of periapical radiolucency	348 (59.5)	339 (97.4%)	0.00
Periapical radiolucency (1–4 mm)	203 (34.7%)	175 (86.2%)	
Periapical radiolucency (>4 mm)	34 (5.8%)	16 (47.1%)	
No walls lost	216 (36.9%)	195 (90.3%)	0.96
One wall lost	222 (37.9%)	201 (90.5%	
Two or more walls lost	147 (25.1%)	134 (91.2%	
Absence of pain	218(37.3%)	190 (87.2%)	0.04
Prolonged spontaneous or stimuli-induced pain	367 (62.2%)	208 (94.1%)	
Pain upon percussion	146 (24.9%)	132 (90.4)	
Treatment cause: caries, pain, or inflammation	499 (85.3%)	463 (92.8%)	0.00
Other causes	86 (14.7%)	67 (77.9%)	
No inflammatory abscess or sinus tract	484 (82.7%)	443 (91.5)	0.07
Inflammation, abscess, or sinus tract	101 (17.3)	87 (86.1%)	
<4 mm probing depth	480 (82.1%)	446 (92.9%)	0.00
>4 mm probing depth	105 (17.9%)	84 (80%)	
Absence of mobility	523 (89.4%)	480 (91.8%)	0.00
Slight mobility	62 (10.6%)	50 (80.6%)	

Absolute values and percentages of teeth which underwent root canal treatment (RCT). Absolute values and percentages of teeth with the absence of signs and symptoms of periapical periodontitis at the end of the follow-up period (complete healing).

**Table 2 jcm-09-01994-t002:** Registered intraoperative and postoperative factors.

**Intraoperative Factors**	**Teeth with RCT**	**Complete Healing**	***p***
Manual instrumentation	118 (20.2%)	103 (87.3%)	0.11
Rotatory instrumentation	467 (79.8%)	427 (91.4%)	
Filling to 0–2 mm of the radiographic apex	410 (70.1%)	373 (91%)	0.36
Infra/over filling	175 (29.9%)	157 (89.7%)	
One to two treatment sessions	471 (80.5%)	429 (91.1%	0.25
More than two treatment sessions	114 (19.5%)	101 (88.6%)	
No intraoperative complications	545 (93.2%)	493 (90.5%)	0.48
Intraoperative complications	40 (6.8%)	37 (92.5%)	
**Postoperative Factors**	**Teeth with RCT**	**Complete Healing**	***p***
Direct composite restoration	493 (84.3%)	459 (93.1%)	0.00
Crown	44 (7.5%)	33 (75.1%)	
Prosthetic bridge	48 (8.2%)	38 (79.2%)	
Presence of antagonist tooth	524 (89.6%)	474 (90.5%)	0.47
Absence of antagonist tooth	61 (10.4%)	56 (91.8%)	
Favorable restoration condition at follow-up	551 (94.2%)	507 (92%)	0.00
Maladjusted or leaked restoration	34 (5.8%)	23 (67.6%)	
1–2 years follow-up	350 (91.1%)	350 (91.1%)	0.48
>2–4 years follow-up	130 (90.9%)	130 (90.9%)	
>4–6 years follow-up	50 (86.2%)	50 (86.2%)	

Absolute values and percentages of teeth which underwent RCT. Absolute values and percentages of teeth with the absence of signs and symptoms of periapical periodontitis at the end of the follow-up period (complete healing).

**Table 3 jcm-09-01994-t003:** Absolute values and percentages of teeth which presented clinical or radiographic factors at the follow-up visits.

Follow-Up Factors	Teeth with RCT
Absence of pain	568 (97%)
Spontaneous pain upon percussion, palpation, or mastication	17 (3%)
No inflammatory abscess or sinus tract	582 (99.5%)
Inflammation, abscess, or sinus tract	3 (0.5%)
<4 mm probing depth	572 (97.8%)
>4 mm probing depth	13 (2.2%)
Absence of mobility	537 (91.8%)
Slight mobility	48 (8.2%)
Absence of periapical radiolucency	530 (90.6%)
Presence of periapical radiolucency with minor or no reduction in size	39 (6.7%)
Presence of periapical radiolucency with an increased size	7 (1.2%)
Presence of a new periapical lesion	9 (1.5%)
Favorable restoration condition at follow-up	551 (94.2%)
Maladjusted or leaked restoration	34 (5.8%)

**Table 4 jcm-09-01994-t004:** Logistic regression analysis.

		95% C.I. for EXP(B)
B	SE	Wald	Gl	Sig.	Exp(B)	Inferior	Superior
Absence of periapical radiolucency			52.062	2	0.000			
1–4 mm periapical radiolucency	−3.766	0.522	52.052	1	0.000	0.023	0.008	0.064
>4 mm periapical radiolucency	−2.167	0.456	22.557	1	0.000	0.115	0.047	0.280
<4 mm probing depth	−1.270	0.371	11.681	1	0.001	0.281	0.136	0.582
Treatment cause: caries, pain, abscess, or sinus tract	−0.839	0.411	4.172	1	0.041	0.432	0.193	0.967
Direct composite restoration			8.616	2	0.013			
Crown	−1.060	0.500	4.496	1	0.034	0.346	0.130	0.923
Prosthetic bridge	0.164	0.654	0.063	1	0.802	1.178	0.327	4.242
Favorable restoration condition at follow-up	−1.942	0.505	14.806	1	0.000	0.143	0.053	0.386
Constant	4.348	0.908	22.950	1	0.000	77.315

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
