# Peer review of "Outcome of Root Canal Treatments Provided by Endodontic Postgraduate Students. A Retrospective Study"

_jcm, 2020, doi:10.3390/jcm9061994_

Round 1

Reviewer 1 Report

Overall comments on MS:

1. The content is of interest to clinicians and may help them in their case selection. 

2. Could you please spell out your abbreviations (ASA and RCT) once, at first mention, both in the abstract and main text. This is important because for instance a dentist will recognize 'RCT' as root canal treatment, but a medical practitioner may think it is randomized clinical trial. 

3. Some grammar needs to be corrected. A. At L 85 it reads as if the clinician specialized in endodontic practice was 18-81 years, when it was actually the patient. B. There were instances where the indirect article 'a' should have been omitted. An example of this is at L 245 'teeth which showed a complete healing' should be ' teeth which showed complete healing'. Please have the article checked again for grammar errors.

Abstract: The dual mention of preoperative and post operative factors  results in repetition and reader confusion, especially when the abstract is read in isolation ( often the case).  Presumably the split was because the first mention was from the chi-squared analysis and the second from the logistic regression. Please rewrite this section to provide more clarity. You could for instance consider just having one lot of preoperative factors and one lot of post operative factors, with each section combining the chi-squared and regression information.

Introduction: 

1. Could you please change the reference at L66 from text format to number format.

2. Suggestion: Consider elaborating on how your paper is different to others.

Experimental Section, Section 2: 

1. This was well explained in general.

2. Could you please mention the radiography technique in this section. Later in the follow up section all that is then required is that the same (if this was the case) technique was used. Because radiography is central to the accuracy of the paper, it is important to state that the clinical and follow up techniques were the same. If they were not, the validity of the paper is then questionable.

Section 3, Clinical follow up evaluation, healing criteria, statistics:

1. Please consider combining sections 2 and 3 into one methodology section with different subsections. This would be more logical. 

2. In Table 1 (third last factor) and Table 3 (second factor) "No inflammatory signs" needs to to changed to "No abscess or sinus tract" or " No inflammatory abscess or sinus tract". This is because pain is also an inflammatory sign, and although pain has been separated out, strictly speaking just saying "no inflammatory sign' could be construed as meaning 'no pain'.

Table 2. You could fill in the empty box in the last factor with the information from your flow chart. Was there a reason that it was left blank?

Results:

1. Similar to the comment in Section 3, in the "Treatment cause section" of Table 4, "inflammation" needs to say "abscess or sinus tract".

2. In Table 4, why do some boxes have no information?  What is the last factor, "Constant"?

Discussion:

1. The discussion was generally comprehensive and well reasoned. 

2. At L 399 presumably you mean intraoperative not preoperative. If this is the case, please fix. 

3. At L 440 you say that some authors find higher success rates when teeth are restored postoperatively with crown and support this with 2 references. However, you go on to say that other authors did not find such an association. Could you please provide a quote to support this statement? If you can't find a quote you will need to reword this statement.

4. In your study extracted teeth could not be included. Hence, teeth that were extracted because of periapical pathology associated with root fracture because of a lack of a crown or those extracted because they developed periapical pathology due to a leaky or broken composite restoration could not be included. I suggest that a comment in his regard may help explain your observations.

Author Response

Thank you very much for your comments, they will surely improve the paper.

  1. The content is of interest to clinicians and may help them in their case selection.

Thank you very much.

  1. Could you please spell out your abbreviations (ASA and RCT) once, at first mention, both in the abstract and main text. This is important because for instance a dentist will recognize 'RCT' as root canal treatment, but a medical practitioner may think it is randomized clinical trial.

Both ASA and RCT have been defined at first mention.

  1. Some grammar needs to be corrected. A. At L 85 it reads as if the clinician specialized in endodontic practice was 18-81 years, when it was actually the patient. B. There were instances where the indirect article 'a' should have been omitted. An example of this is at L 245 'teeth which showed a complete healing' should be ' teeth which showed complete healing'. Please have the article checked again for grammar errors.

Grammar has been revised and corrected throughout the text.

Abstract: The dual mention of preoperative and post operative factors results in repetition and reader confusion, especially when the abstract is read in isolation (often the case).  Presumably the split was because the first mention was from the chi-squared analysis and the second from the logistic regression. Please rewrite this section to provide more clarity. You could for instance consider just having one lot of preoperative factors and one lot of post operative factors, with each section combining the chi-squared and regression information.

The abstract has been rewritten following your recommendations.

Introduction:

  1. Could you please change the reference at L66 from text format to number format.

The reference format has been changed.

  1. Suggestion: Consider elaborating on how your paper is different to others.

A comment was added in the discussion section highlighting this.

Experimental Section, Section 2:

  1. This was well explained in general.
  2. Could you please mention the radiography technique in this section. Later in the follow up section all that is then required is that the same (if this was the case) technique was used. Because radiography is central to the accuracy of the paper, it is important to state that the clinical and follow up techniques were the same. If they were not, the validity of the paper is then questionable.

The radiographic technique has been described in this section.

Section 3, Clinical follow up evaluation, healing criteria, statistics:

  1. Please consider combining sections 2 and 3 into one methodology section with different subsections. This would be more logical.

Sections 2 and 3 have been combined.

  1. In Table 1 (third last factor) and Table 3 (second factor) "No inflammatory signs" needs to to changed to "No abscess or sinus tract" or " No inflammatory abscess or sinus tract". This is because pain is also an inflammatory sign, and although pain has been separated out, strictly speaking just saying "no inflammatory sign' could be construed as meaning 'no pain'.

Texts in tables 1 and 3 have been changed.

Table 2. You could fill in the empty box in the last factor with the information from your flow chart. Was there a reason that it was left blank?

The boxes were empty because values are the same as in those in the third column. Now we have added these values.

Results:

  1. Similar to the comment in Section 3, in the "Treatment cause section" of Table 4, "inflammation" needs to say "abscess or sinus tract".

The text in table 4 has been changed.

  1. In Table 4, why do some boxes have no information? What is the last factor, "Constant"?

In a logistic regression model, the variable has three possible categories or values. The reference one is compared with the others (the first with the second and the first with the third one), then, the first box is blank, it has no values.

A constant is always present in a logistic regression model. It appears as a consequence of the model’s settlement .

Discussion:

  1. The discussion was generally comprehensive and well reasoned.
  2. At L 399 presumably you mean intraoperative not preoperative. If this is the case, please fix.

It was a mistake. It has been corrected.

  1. At L 440 you say that some authors find higher success rates when teeth are restored postoperatively with crown and support this with 2 references. However, you go on to say that other authors did not find such an association. Could you please provide a quote to support this statement? If you can't find a quote you will need to reword this statement.

A reference has been added.

  1. In your study extracted teeth could not be included. Hence, teeth that were extracted because of periapical pathology associated with root fracture because of a lack of a crown or those extracted because they developed periapical pathology due to a leaky or broken composite restoration could not be included. I suggest that a comment in his regard may help explain your observations.

A new phrase has been added in the discussion section regarding your comment. The decision to not include extracted teeth came from the impossibility to evaluate them on the follow-up visits.

Reviewer 2 Report

Review of “Outcome of Root Canal Treatments Provided by Endodontic Postgraduate Students. A Retrospective Study.”

I would like to congratulate the authors for presenting a manuscript that brings some insights on factors possibly affecting healing of periapical lesions after primary root canal treatment. Notwithstanding that, there are some issues that need to be clarified.

  1. Why were teeth grouped as anterior teeth and premolars (table 1)? Premolars cannot be considered monoradicular teeth and therefore the associated risk cannot be similar.
  2. Please do not use p=0.00 values. Use a different denomination as p<0.001.
  3. The population should be described prior to the tables. Mean age, mean time elapsed from RCT and follow-up, gender frequency, diagnosis prior to RCT, open and closed apex, among others – all this initially without considering the outcome and only then introducing that.
  4. The rotary files systems should be described, as well as the conditions of the canal preparation: RPMs, torque, use of lubricants, etc
  5. Please also detail the operative complications: broken files, hypochlorite accidents, contamination, etc
  6. Why are there results in the experimental section / materials and methods? The manuscript numbering is not correct. Please consider reordering your manuscript.
  7. Kappa values are not percentages. Why are there 4 values of kappa agreement?
  8. The logistic regression model should be more carefully presented and explained – risks associated with each factor.

Author Response

Review of “Outcome of Root Canal Treatments Provided by Endodontic Postgraduate Students. A Retrospective Study.”

I would like to congratulate the authors for presenting a manuscript that brings some insights on factors possibly affecting healing of periapical lesions after primary root canal treatment. Notwithstanding that, there are some issues that need to be clarified.

Thank you very much for your comments that will surely improve the text.

Why were teeth grouped as anterior teeth and premolars (table 1)? Premolars cannot be considered monoradicular teeth and therefore the associated risk cannot be similar.

We considered three groups of teeth according to the difficulty of the case, following the classification from the American Association of Endodontists (AAE). We have added this information in the text (discussion section).

Please do not use p=0.00 values. Use a different denomination as p<0.001.

P=0.00 has been changed by p<0.001.

The population should be described prior to the tables. Mean age, mean time elapsed from RCT and follow-up, gender frequency, diagnosis prior to RCT, open and closed apex, among others – all this initially without considering the outcome and only then introducing that.

A new paragraph has been included at the beginning of the results section, describing, basically, the study population. This is followed by more detailed already present information. Tables have been reordered and placed in the results section, as suggested by the reviewer, in a complementary way between text and tables.

The rotary files systems should be described, as well as the conditions of the canal preparation: RPMs, torque, use of lubricants, etc

As explained in the text, different manual and rotary instruments were used based on the operator´s preferences (always professor-controlled), even alternating different systems, as we do not use only one system in our postgraduate program in endodontics. Irrigation procedures are explained in the text.

Please also detail the operative complications: broken files, hypochlorite accidents, contamination, etc.

There were a 6.8% of intraoperative complications. This information is included in table 2.

Why are there results in the experimental section / materials and methods? The manuscript numbering is not correct. Please consider reordering your manuscript.

Following the previous comment of the reviewer, we have reordered the manuscript, changing the placement of the tables, according to the text.

Kappa values are not percentages. Why are there 4 values of kappa agreement?

You are right. Percentage symbols were a mistake, they have been deleted.  As there were two observers, there are two intra observer Kappa indexes, but only one inter observer kappa value. All this has been changed in the text.

The logistic regression model should be more carefully presented and explained – risks associated with each factor.

We have added a paragraph explaining the clinical relevance of the logistic regression analysis, including an example. Complete information is shown in table 4.

Round 2

Reviewer 2 Report

Thank you for the improved manuscript. I feel it is OK to be published. 

Author Response

Reviewer response

“Thank you for the improved manuscript. I feel it is OK to be published.”

Thank you for your comments. Following your review report form, the manuscript was proofread and a series of minor corrections were made in terms of the English language and grammar. Additionally, the conclusion was rewritten in order to improve its consistency with the results.
